# Properties and Functional Analysis of Two Chorismate Mutases from Maritime Pine

**DOI:** 10.3390/cells13110929

**Published:** 2024-05-28

**Authors:** Fernando de la Torre, Beatriz Medina-Morales, Irene Blanca-Reyes, M. Belén Pascual, Concepción Ávila, Francisco M. Cánovas, Vanessa Castro-Rodríguez

**Affiliations:** Departamento de Biología Molecular y Bioquímica, Facultad de Ciencias, Universidad de Málaga, Campus Universitario de Teatinos, 29071 Málaga, Spain; beatrizmedinamorales96@gmail.com (B.M.-M.); ireneblanca@uma.es (I.B.-R.); bpascual@uma.es (M.B.P.); cavila@uma.es (C.Á.); canovas@uma.es (F.M.C.)

**Keywords:** chorismate, chorismate mutase, chloroplast, aromatic amino acids, shikimate pathway, conifers

## Abstract

Through the shikimate pathway, a massive metabolic flux connects the central carbon metabolism with the synthesis of chorismate, the common precursor of the aromatic amino acids phenylalanine, tyrosine, and tryptophan, as well as other compounds, including salicylate or folate. The alternative metabolic channeling of chorismate involves a key branch-point, finely regulated by aromatic amino acid levels. Chorismate mutase catalyzes the conversion of chorismate to prephenate, a precursor of phenylalanine and tyrosine and thus a vast repertoire of fundamental derived compounds, such as flavonoids or lignin. The regulation of this enzyme has been addressed in several plant species, but no study has included conifers or other gymnosperms, despite the importance of the phenolic metabolism for these plants in processes such as lignification and wood formation. Here, we show that maritime pine (*Pinus pinaster* Aiton) has two genes that encode for chorismate mutase, *PpCM1* and *PpCM2*. Our investigations reveal that these genes encode plastidial isoenzymes displaying activities enhanced by tryptophan and repressed by phenylalanine and tyrosine. Using phylogenetic studies, we have provided new insights into the possible evolutionary origin of the cytosolic chorismate mutases in angiosperms involved in the synthesis of phenylalanine outside the plastid. Studies based on different platforms of gene expression and co-expression analysis have allowed us to propose that PpCM2 plays a central role in the phenylalanine synthesis pathway associated with lignification.

## 1. Introduction

The shikimate pathway is used by bacteria, archaea, fungi, algae, some protozoans, and plants for the biosynthesis of the aromatic amino acids phenylalanine (Phe), tyrosine (Tyr), and tryptophan (Trp), but also of a vast array of phenolic-derived compounds [1]. This pathway exhibits one of the largest metabolic fluxes in the biosphere, and in plants, it is estimated that this route channels up to 30% of the photosynthetically fixed C [2]. Starting from the central C metabolites phosphoenolpyruvate and erythrose-4-phosphate within plastids, the shikimate pathway results in the synthesis of chorismate, the last common precursor for the three aromatic amino acids, as well as other compounds of paramount importance, such as vitamins K1 and B9. Chorismate is mainly used as a substrate by the enzyme anthranilate synthase (AS; EC 4.1.3.27), for the synthesis of tryptophan, or by the enzyme chorismate mutase (CM; EC 5.4.99.5) for the synthesis of prephenate, the common precursor of Phe and Tyr. The alternative utilization of chorismate by these two enzymes is finely regulated by intracellular levels of aromatic amino acids and, in some cases, by other amino acids or downstream metabolites. In plants, tryptophan generally functions as an inhibitor of the AS enzyme and activator of CM [3,4]. Conversely, phenylalanine and tyrosine function as inhibitors of CM activity and activators of the AS [5]. Additionally, a characteristic feature of angiosperms is the existence of cytosolic CMs involved in the extra-plastidial synthesis of Phe, which are insensitive to regulation by amino acids [4,6].

Previous works have addressed the phylogenetic distribution of CMs, showing the existence of several clades that partially correlate with their allosteric regulation. Briefly, these studies establish the existence of a distinct group that includes CMs from green algae, a clade corresponding to the cytosolic enzymes from angiosperms, as well as another clade containing the angiosperm plastidic isoforms [5,7]. Finally, an additional clade encompassing CMs from basal plant lineages, such as lycophytes or mosses, was also reported [5]. None of the aforementioned analyses considered the inclusion of CMs from conifers or other gymnosperms, likely due to the delayed completion of sequencing projects for these plant species with extremely complex and repetitive megagenomes.

Parallel studies have also explored structural aspects of plant CMs, focusing particularly on the residues involved in the binding of amino acid effectors. For instance, Westfall et al. (2014) [3] utilized 3D modeling and site-directed mutagenesis to demonstrate that in AtCM1, Gly213 is essential for allosteric regulation, while Gly149 contributes to determining the specificity of the effector amino acids. The importance of these analyses is highlighted by the conservation of Gly213 in CMs within the phylogenetic clade encompassing plastidial regulated CMs, in contrast with its absence in the clade of cytosolic CMs which are insensitive to inhibition by Phe or Tyr.

Conifers, owing to their significant vertical growth and exposure to multiple mechanical stresses, require substantial lignin synthesis, and therefore, large amounts of its precursor, phenylalanine. In previous work, we studied the biochemistry and molecular regulation of this crucial process, using *Pinus pinaster* (maritime pine) as a model species. In particular, we extensively examined the enzymes prephenate aminotransferase and arogenate dehydratase, which are responsible for synthesizing Phe from prephenate, the product of the reaction catalyzed by CM [8,9,10,11]. Additionally, we contributed to elucidating a regulation model of the Phe synthesis pathway, involving MYB and NAC type transcription factors [12,13,14]. 

The overarching objective of this work was to investigate the role of the CM enzyme in controlling the metabolic flux of phenolic compounds in conifers, a plant group that is reliant on substantial synthesis of these compounds. In this regard, here we have analyzed the CM family in maritime pine, comprising two members with plastidial localization, reinforcing the absence of a cytosolic Phe synthesis pathway in conifers. Furthermore, we investigated the biochemical regulation of these enzymes and elucidated how this metabolic step correlates at the transcriptional level with other key steps of this pathway.

## 2. Methods

### 2.1. Growth Conditions

*Nicotiana benthamiana* seeds were planted and allowed to grow for 6 weeks. Initially, seeds were planted in pots for 1–2 weeks (depending on the size), then each seedling was separated into individual pots for 3–4 weeks. *N. benthamiana* seedlings were grown at 25 °C in a plant chamber with a long-day photoperiod (16 h of light and 8 h of darkness).

Seeds from maritime pine were soaked in distilled water for 24 h with continuous aeration, then germinated in a plastic tray filled with vermiculite as the substrate. The seedlings were grown in a controlled environment growth room, maintained at 25 °C and 50/70% relative humidity, with a 16/8 h photoperiod, and were watered twice weekly with distilled water.

### 2.2. Maximum Likelihood Phylogeny Analysis

CM phylogenetic analysis was conducted with 135 sequences corresponding to multiple species from different plant clades and green algae. The CM from *Saccharomyces cerevisiae* was used to outgroup root the tree. Sequences were aligned using muscle [15] and tree topology was inferred using maximum likelihood with PhyML [16,17]. Bootstrapping was performed with 1000 replicates. The phylogenetic trees were drawn using MEGA 11 [18].

### 2.3. cDNA Cloning

Full-length cDNAs of *PpCM1* to *PpCM2* were obtained from *Pinus pinaster* seedlings’ total RNA through reverse transcription-PCR. This process involved the use of iScript Reverse Transcription Supermix (Bio-Rad, Hercules, CA, USA) and the corresponding primer pairs listed in Appendix A. Subsequently, the obtained cDNAs were subcloned into the pJET1.2 vector (ThermoFisher Scientific, Waltham, MA, USA). Sequenced full-length cDNAs were PCR amplified using forward and reverse oligonucleotides that featured attB1 and attB2 sites at their 5’ and 3’ ends. These primer pairs are all detailed in Appendix A. The resulting products were cloned into the pDONR207 vector, and then into pDEST17 vector, using the BP Clonase^®^ II and LR Clonase^®^ II enzyme mixes (Thermo Fisher Scientific, Waltham, MA, USA), respectively. 

### 2.4. RNA Extraction, Reverse Transcription, and Quantitative Real-Time PCR

#### 2.4.1. RNA Extraction

Samples from developing xylem from compressed and opposite woods were obtained, as described by Villalobos et al., 2012 [19]. Samples were processed to extract total RNA in accordance with the procedure by Canales et al., 2012 [20]. Approximately 100 mg of each powdered sample was employed in the extraction process. To eliminate any genomic DNA contaminants from the RNA samples, 5 units of RQ1 Rnase-Free Dnase (Promega, Madison, WI, USA) were applied for 25 min at 37 °C. The purity of the total RNA, as indicated by the 260/280 and 260/230 ratios, was assessed using a NanoDrop ND-1000 spectrophotometer (ThermoFisher Scientific, Waltham, MA, USA), which was also employed to quantify the samples. Furthermore, the quality of the total RNA was evaluated through agarose gel electrophoresis.

#### 2.4.2. cDNA Synthesis and qPCR Analyses

For cDNA synthesis, 0.5 μg of total RNA was utilized in accordance with the manufacturer’s guidelines, employing the qPCRBIO cDNA synthesis kit (PCR Biosystems, London, UK). The qPCR analysis involved the utilization of specific primers, which can be found in Appendix A. The qPCR reaction mixture was prepared as follows: 5 μL of 2 × SsoFast™ EvaGreen^®^ Supermix (BioRad, Hercules, CA, USA), 0.5 pmol of the forward primer, 0.5 pmol of the reverse primer, and 10 ng of the amplified cDNA. The qPCR reactions were conducted in a CFX384 thermal cycler (Bio-Rad, Hercules, CA, USA) under the following conditions: an initial denaturation step of 3 min at 95 °C (1 cycle), followed by a 1 s denaturation at 95 °C, and a 5 s annealing/extension at 60 °C (for 50 cycles). A subsequent melting curve analysis was performed over a temperature range from 60 °C to 95 °C.

The raw fluorescence data from each reaction were analyzed using the MAK2 model, which does not assume a specific amplification efficiency for quantitative PCR (qPCR) assays, as described by Boggy and Woolf, 2008 [21]. The initial target concentration (D0 parameter) for each gene was determined using the MAK2 model and the qPCR package in the R environment, as outlined by Ritz and Spiess, 2008 [22]. Subsequently, these values were normalized to two reference genes, *PpActin* and *PpEF1* (Appendix A).

### 2.5. Transcriptome Libraries

In silico expressions of *PpCMs* were studied using different transcriptome databases. These include Sustainpine seedlings (http://v22.popgenie.org/microdisection/, accessed on 15 April 2024) [23] and developing xylem [19] databases.

### 2.6. Subcellular Localization of CMs

Pine *PpCMs* were cloned into the pGWB5 vector using gateway technologyto create complete PpCM proteins tagged with GFP at their C-termini, under the regulation of the CaMV 35S promoter. The resulting plasmids were introduced into *Agrobacterium tumefaciens* C58C1 strains via electroporation. The expression of GFP-tagged PpCM proteins was achieved through agroinfiltration with an OD_600_ of 0.5 in *Nicotiana benthamiana* leaves, following established procedures [24]. In all experiments, the silencing suppressor p19 [25] was co-expressed. GFP fluorescence was assessed 48 h post-agroinfiltration using a Leica Stellaris 8 confocal microscope (Leica, Wetzlar, Germany), for chloroplast autofluorescence of excitation/emission at 488/680–700 nm and for GFP detection excitation/emission at 488/505–525 nm. Excitation was provided by an argon ion laser 488 nm. The expression of the corresponding proteins was confirmed by western blot.

### 2.7. Protein Extraction, SDS-PAGE, and Immunodetection

Total proteins were extracted from plant material using an extraction buffer (100 mM Tris buffer at pH 8.0, 10% (*v*/*v*) glycerol, 1% (*w*/*v*) sodium dodecyl sulfate (SDS), 2 mM EDTA, and 0.1% (*v*/*v*) beta-mercaptoethanol). Approximately 100 mg of frozen plant powder was reconstituted in 150 μL of the extraction buffer at room temperature. Intact chloroplast isolation was performed using a CPISO chloroplast isolation kit (Merck, Darmstadt, Germany) and a Percoll^®^ gradient.

After centrifugation at 20,000× *g* for 10 min at 4 °C, 75 μL of the supernatant was collected and mixed with 25 μL of 4× Laemmli buffer, followed by denaturation at 100 °C for 5 min. 

For the immunodetection of transiently expressed CMs within the protein extracts, 25 μg of the total proteins was separated using sodium dodecyl sulfate polyacrylamide gel electrophoresis (SDS-PAGE). Western blot analysis was conducted following standard procedures. Transgenic proteins were detected by exploiting the GFP tag present in the construct, utilizing a specific commercial antibody (GFP (B-2) sc-9996, mouse monoclonal antibody, Santa Cruz Biotechnology, Santa Cruz, CA, USA) at a dilution of 1:1000.

### 2.8. Recombinant Expression and Purification of CM Enzymes in Escherichia coli

DNA sequences coding for both PpCM1 and PpCM2 were subcloned into the pDEST17 vector using Gateway technology. In both cases, the sequence coding for the putative chloroplast transit peptide was not included. These sequences, coding for residues 1–58 of PpCM1 and residues 1-58 of PpCM2, were determined by comparison with multiple CM sequences from plants, using ChloroP [26] and TargetP [27] algorithms.

pDEST17 expression constructs were transformed into *Escherichia coli* BL21-CodonPlus-RIL^®^ (Agilent Technologies, Santa Clara, CA, USA) and cultured in 500 mL of LB, supplemented with 100 µg of ampicillin and 34 µg of chloramphenicol, at 37 °C until *A*_600_ of 0.6 was reached. Protein expression was induced by adding a final concentration of 1 mM isopropyl β-D-1-thiogalactopyranoside (IPTG) to the cultures, which were subsequently incubated for 5 h at 20 °C with gentle shaking at 100 rpm. Cells were pelleted by centrifugation at 6000× *g* and frozen until further purification. Poly-His-tagged recombinant proteins were purified using Protino Ni-TED 2000 Packed nickel resin columns (Macherey-Nagel, Duren, Germany) and subjected to buffer exchange to a 50 mM Tris buffer at pH 8.0 using Sephadex G-25 M resin (PD-10 Columns; GE Healthcare, Chicago, IL, USA).

### 2.9. Co-Immunoprecipitation Analysis

DNA sequences coding for both PpCM1 and PpCM2 were subcloned into the pGWB11 (C-terminal FLAG tag) and pGWB5 (C-terminal GFP tag) vectors, respectively, using Gateway technology. PpCM2-GFP and PpCM1-FLAG proteins were expressed through agroinfiltration in *Nicotiana benthamiana* leaves, as described in Section 2.6. The empty vector pGWB6 (N-terminal GFP tag) was used as a control.

Co-immunoprecipitation was performed using ChromoTek GFP-Trap^®^ Agarose (Chromotek, Planegg-Martinsried, Germany). Briefly, soluble proteins were extracted from 250 mg of *N. benthamiana* leaves in an extraction buffer containing 50 mM Tris-HCl, pH 7.5; 150 mM NaCl; 10% glycerol; 2 mM EDTA, pH 8; 1% Triton X-100, and 1% protease inhibitor cocktail P-9599 from Merck KGaA (Darmstadt, Germany). After centrifugation at 20,000× *g* for 15 min at 4 °C, 100 µL was saved as input for western blot analysis and remaining supernatants (two milligrams of total protein extracts) were mixed with 25 μL of GFP-Trap A beads and incubated 2 h at 4 °C, with end-over-end rocking. After incubation, the beads were washed three times with wash buffer (extraction buffer). Proteins bound to the beads were resuspended in 80 mL of 2× Laemmli sample buffer and heated at 100 °C for 10 min to dissociate immunocomplexes from the beads. Total (input) immunoprecipitated (IP) and co-immunoprecipitated (CoIP) proteins were separated by electrophoresis in 10% SDS-PAGE and analyzed by western blot analysis using anti-Flag (1:1000, OctA-Probe H5, sc-166355, Santa Cruz Biotechnology, Santa Cruz, CA, USA) or anti-GFP (1:1000; sc-9996, Santa Cruz Biotechnology, Santa Cruz, CA, USA) antibodies. Appropriate peroxidase-conjugated secondary antibodies were used: m-IgGk BP-HRP; sc-516102 (Santa Cruz Biotechnology, Santa Cruz, CA, USA) for anti-Flag (1:5000) and anti-mouse igG-Peroxidase, A9044 from Merck Darmstadt, Germany for anti-GFP (1:5000).

### 2.10. CM Activity Assay

CM reactions consisted of a final volume of 80 µL containing 50 mM Tris buffer pH 8, varying concentrations of chorismate, and 0.3 µg of the corresponding purified CM enzyme. CM assays were conducted in a plate reader at 30 °C, using UV-transparent 96-well plates (UV-Star^®^, Greiner Bio-One. GmbH, Frickenhausen, Germany), by tracking the disappearance of chorismate, which results in an absorbance decrease at A_274 nm_ (ε = 2630 M^−1^ cm^−1^). Chorismate (C1259) was obtained from Merck KGaA (Darmstadt, Germany) To identify putative amino acid effectors, the reactions were conducted as described above; containing 50 mM Tris buffer pH 8, 1 mM chorismite, 0.3 µg of the CM enzyme, individually supplemented with 2 mM of each amino acid. Assays showing a minimum 1.5-fold increase or reduction compared to controls were selected for further analysis. Similarly, the effect on CM activity of decreasing concentrations of the amino acids Phe, Tyr, and Trp was determined. 

## 3. Results

### 3.1. The CM Protein Family of Pinus pinaster: Structure and Phylogenetic Relationships

Based on the *P. pinaster* databases, we identified two members in the CM family of *Pinus pinaster*, which we will refer to as PpCM1 and PpCM2 henceforth. Both sequences present putative chloroplast transit peptides and show amino acid identity close to 67%. The genetic structure of the CM family of *P. pinaster* aligns well with the findings in the present study of two-members in other gymnosperm species, such as *Gnetum montanum*, *Ginkgo biloba*, *Pinus taeda*, *Picea abies* or *Thuja plicata* (Appendix A). Additionally, our investigation extended to CM sequences across various plant groups, including green algae, mosses, ferns, lycophytes, and angiosperms. Across most species included here the number of members in the CM family ranged between two and five, with green algae being the notable exception, displaying only a single CM (Appendix A).

Previous phylogenetic analyses have revealed distinct clades for the CM enzyme in plants [3]. To identify the phylogenetic distribution of the CM proteins of *Pinus pinaster* and globally in conifers, we conducted a phylogenetic analysis of CM protein sequences from various photosynthetic organisms, including chlorophytes, lycophytes, ferns, mosses, liverworts, gymnosperms, and angiosperms, including basal lineages (Figure 1 and Appendix A). These analyses indicated that CMs from Viridiplantae mainly cluster into three main clades. Clade-I includes sequences from chlorophytes, along with the sequence of the yeast *Saccharomyces cerevisiae,* and the sequence of the Rhodophyta *Porphyra umbilicalis* (Figure 1). Clade-II includes plastid CMs from all plant lineages, including a basal sub-clade exclusive to ancient groups like lycophytes, ferns, mosses, and liverworts (Figure 1). This clade also contains sequences corresponding to CMs from various gymnosperm species, such as pine, *Ginkgo biloba,* or *Gnetum montanum*, including PpCM2 (Figure 1). Clade-III predominantly consists of the cytosolic, allosterically deregulated isoforms from angiosperms, but also includes a subgroup of plastidial CMs from gymnosperms, such as PpCM1 (Figure 1). Interestingly, *Amborella trichopoda*, belonging to the most basal lineage of extant angiosperms [28], also displays two plastidial isoforms distributed across the second and third clade. Conversely, other basal angiosperms and Magnoliids like *Nymphaea colorata*, *Cinnamomum kanehirae,* or *Liriodendron tulipifera,* like most angiosperms, possess both cytosolic and plastidial enzymes distributed into clades -II and -III, respectively (Figure 1).

### 3.2. PpCMs Are Localized in the Plastid Stroma

Previous studies on chorismate mutases in plants revealed that variants could exhibit plastidic or cytosolic localization. Using the alternative prediction programs ChloroP and TargetP, and through sequence comparison (Appendix A), the existence of putative N-terminal transit peptides for plastid-targeting in the primary structure of both pine CM isoforms was determined. To confirm their plastid localization, C-terminal GFP fusions for both PpCM1 and PpCM2 were transiently expressed in *N. benthamiana* leaves via agroinfiltration. Within 16–24 h, fluorescent signals were detected only in chloroplasts (Figure 2A,B). To further validate these observations, chloroplasts were isolated by differential centrifugation and rapid purification on Percoll^®^ gradients from *N. benthamiana* disc leaves alternatively expressing PpCM1-GFP, PpCM2-GFP, and PpADT-A, with the last one included as a chloroplast protein control [10]. The presence of the corresponding fusion proteins in total proteins from disc leaves (CE), but also in soluble (S1) and insoluble (S2) proteins from chloroplasts, was analyzed by western blotting. As shown in Figure 2B, the relative abundance of the polypeptides PpCM1-GFP and PpCM2-GFP was enriched in the lanes containing soluble chloroplast proteins (S1), which also occurred with the chloroplast-located enzyme PpADT-A [10]. Based on these results, we concluded that both PpCM1 and PpCM2 are located in the plastid stroma.

### 3.3. The Spatial Expression Patterns of PpCMs Suggest Complementary Roles of the Two Isoforms in Maritime Pine

Using the transcriptome atlas of *Pinus pinaster* seedlings developed in our laboratory, we analyzed the expression patterns corresponding to *PpCM1* and *PpCM2* [23]. The results indicate that *PpCM1* exhibits preferential expression in cotyledon mesophyll cells, hypocotyl pith, and developing root vascular tissues. Interestingly, its expression decreases in the apical meristem, as well as in young needles’ vascular and hypocotyl cortexes (Figure 3). Conversely, *PpCM2* transcripts were more abundant in vascular tissues from young needles, as well as in the vascular cells of cotyledons and hypocotyls (Figure 3). These findings support distinct metabolic functions for PpCM1 and PpCM2, suggesting that they play complementary roles during the initial stages of tree development. 

### 3.4. PpCMs and PpADTA-D Co-Expression Was Specifically Detected in Developing Compression Wood

Conifers form a unique woody tissue, termed compression wood (CW), beneath branches and inclined stems. Simultaneously, these branches and stems undergo the development of (opposite wood) OW [29]. Unlike OW, CW exhibits heightened lignin biosynthesis and deposition, accompanied by a reduction in cellulose content. To obtain further insights into the function of CM genes in maritime pine, their co-expression with several members of the arogenate dehydratase (ADT) family was examined during wood formation. As shown in Figure 4, *PpCM1* exhibits unique interactions with *PpADT-D*, and additionally connects with *PpADT-I* and *PpADT-H*. Interestingly, *PpADT-I* and *PpADT-H* also interact with each other, forming a network, and they also connect with *PpADT-E* and *PpCM2*. On the other hand, *PpCM2* displays exclusive interaction with *PpADT-A*. RNAseq analysis of developing opposite wood and developing compressed wood reveals that both *PpCM1* and *PpCM2* share expression profiles with *PpADT-A* and *PpADT-D*. In a prior study, transcript levels of *PpADT-A* and *PpADT-D* were analyzed, as well as the expression profiles of other members of the pine *ADT* gene family in CW and OW samples obtained from mechanically stressed 25-year-old trees with specialized vascular tissues [14]. In Figure 4C, RT-qPCR confirms the previously observed expression profiles. This validation underscores the reliability of the expression patterns identified in *PpCM1*, *PpCM2*, *PpADT-A*, and *PpADT-D,* in the context of developing opposite wood and developing compressed wood.

### 3.5. Catalytic Activities of PpCMs

To study their catalytic properties, *P. pinaster* PpCM1 and PpCM2 were functionally expressed in *E. coli,* including N-terminal His-tags, lacking the predicted chloroplast signal peptides, and were subsequently purified with Ni^2+^-affinity chromatography. Following this procedure, the steady-state kinetic parameters were determined for these enzymes by measuring the disappearance of chorismate, which leads to an absorbance decrease at 274 nm. Our results indicate that both enzymes have similar turnover rates, 29.4 s^−1^ for PpCM1 and 35 s^−1^ for PpCM2 (Table 1). These values of K_cat_ are in the range of those described for the enzymes of *Arabidopsis,* between 13 and 39 s^−1^ [3], but also in the range of those reported for CMs from ancient plant clades, such as mosses (19.5–20.7 s^−1^), lycophytes (18.8 s^−1^), or basal angiosperms (15–22.8 s^−1^) [30]. Regarding K_m_ values, our results also indicate a similar affinity for the chorismate of PpCM1 (1.6 mM) and PpCM2 (1.7 mM). These values are slightly higher than those reported for *Arabidopsis*, between 0.15 and 1.10 mM [3], and slightly lower than those described for mosses, lycophytes, and *A. trichopoda* [30]. Therefore, the catalytic efficiency values for the alternative CMs from *P. pinaster* are similar and intermediate between the highest values of *Arabidopsis* CMs and the lowest values of CMs from more ancestral groups of plants.

### 3.6. Role of Amino Acids as Effectors on Pine CM Activity

To gain a deeper understanding of the regulation of pine CMs, we investigated the potential role of all 20 proteinogenic amino acids as effectors. The impact of each of these amino acids on CM activity was analyzed at a concentration of 2 mM, and assays resulting in a 1.5-fold impact on activity, either positive or negative, were considered for more detailed study. Only the aromatic amino acids phenylalanine, tyrosine, and tryptophan displayed activation or inhibition above this level (Appendix A). 

Our results indicate that both PpCM1 and PpCM2 are subjected to marked regulation through negative feedback inhibition by phenylalanine and tyrosine. Compared to controls, PpCM1 and PpCM2 showed a 9.6- and 3-fold reduction in activity, respectively, when assayed in presence of 2 mM Phe, with EC_50_ values for PpCM1 (36 µM) lower than those observed for PpCM2 (88 µM) (Table 1). Notably, PpCM2 retained around 30–35% of its activity in the presence of 2 mM of Phe (Figure 5). For tyrosine, the reduction in activity in the presence of 2 mM was comparable for both enzymes, with 11.4- and 10.1-fold decreases for PpCM1 and PpCM2, respectively (Table 1 and Figure 5). The determined EC_50_ value for PpCM1, 9.1 µM, was close to half of that observed for PpCM2, 19 µM (Figure 5).

Regarding tryptophan, both enzymes displayed enhanced activity at 2 mM, with a 3.1-fold increase for PpCM1 and a 1.7-fold increase for PpCM2. Interestingly, both enzymes showed similarly low EC_50_ values close to 0.1 µM, indicating a high sensitivity to low concentrations of Trp. These results suggest that, compared to other vascular plants, pine CMs are much more sensitive to activation through tryptophan, and are capable of maintaining considerable activity enhancement, even at the nanomolar level (Figure 5).

Overall, these results indicate that although both enzymes have similar kinetic parameters and their activity is regulated by the same amino acid effectors; PpCM1 is more sensitive to Phe, Tyr, and Trp than PpCM2, which is phylogenetically closer to the deregulated cytosolic CMs that exist in angiosperms.

### 3.7. Prospective Analysis of Heterodimer Formation between Alternative Maritime Pine CMs

As described above, through heterologous expression in *E. coli,* we have produced and characterized recombinant enzymes corresponding to the two isoforms of pine CMs. As widely reported, CMs from plants, but also CMs from bacteria and yeast, exist as dimeric holoenzymes. Our transcriptional analyses have shown that, in certain tissues and under specific physiological situations, the expression patterns of the two CM genes suggest that the alternative isoforms could coexist in certain plastids. Although, to date, there are no studies addressing this point, we cannot rule out the in vivo formation of heterodimers for these enzymes. To investigate this possibility, we decided to perform a co-immunoprecipitation analysis. PpCM1 protein, with a C-terminal FLAG tag, was transiently co-expressed in *Nicotiana benthamiana* leaves with PpCM2, containing a C-terminal GFP tag (Figure 6). In this experiment, we observed that immunoprecipitation of PpCM2 results in the co-immunoprecipitation of PpCM1, indicating the in vivo formation of a heterodimer, an effect which, to date, has not been detected on CM enzymes (Figure 6).

## 4. Discussion

In our previous studies, we analyzed the regulation of the enzyme prephenate aminotransferase (PAT), which converts prephenate into arogenate, and the enzyme arogenate dehydratase (ADT), which catalyzes the synthesis of Phe from arogenate, in *Pinus pinaster* [8,10,11,14]. This species serves as a well-accepted model for conifers, a group of plants with a complex secondary metabolism requiring a high rate of synthesis of compounds derived from aromatic amino acids to maintain their growth and development. 

Here, we investigate the regulation of the enzyme CM in the same species, with the aim of obtaining a comprehensive understanding of the metabolic regulation of Phe synthesis starting from chorismate. The CM reaction is an essential step that determines the metabolic flux towards the synthesis of Phe and Tyr, as well as multiple derived compounds. This pathway competes with the alternative use of chorismate towards other important metabolic destinations, including the synthesis of Trp and other compounds, such as vitamins K1 and B9 [5]. The alternative channeling of chorismate towards different metabolic destinations represents a branching point, regulated through positive and negative feedback loops, which has been extensively studied in plant CMs, as well as in the enzymes of microorganisms, including bacteria and yeast [5,31]. 

In general, CM enzymes are inhibited by the amino acids derived from prephenate, Phe and Tyr, and activated by Trp. In contrast, the enzyme anthranilate synthase, which catalyzes the first step of Trp synthesis from chorismate, is activated by Phe and Tyr and inhibited by Trp (reviewed by Maeda and Dudareva, 2012 [5]). Several works have delved into this general model, evidencing different types of regulation for the enzyme chorismate mutase in various plant species and cellular compartments. Notably, angiosperms possess cytosolic CM enzymes whose activity, in the cases studied, appears to be insensitive to aromatic amino acids. Regarding plastidial enzymes, alternative regulatory mechanisms exist, as illustrated by the case of *Arabidopsis*, in which the AtCM1 enzyme is positively activated by Trp and negatively regulated by Tyr and Phe, while the AtCM3 enzyme is activated by Trp, His, and Cys [3]. Although these processes mainly occur in chloroplasts, recent works have shown that in some plant groups, the synthesis of aromatic amino acids can also partially take place in the cytosol [32].

In this work, we report the existence of two coding sequences for the enzyme chorismate mutase, PpCM1 and PpCM2, in *Pinus pinaster*. In parallel, we also identified CM sequences in other species of conifers and gymnosperms, which were also found to contain two *CM* genes in their genomes. Through sequence alignments, in silico analysis, and transient expression studies in *Nicotiana benthamiana* leaves of the fusion proteins PpCM1-GFP and PpCM2-GFP, we concluded that both isoforms are specifically located in plastids, indicating that, unlike angiosperms, *Pinus pinaster*, and probably all conifers, lack cytosolic CM enzymes. This finding is of particular importance, since the cytosolic pathway for Phe synthesis described in petunia, and proposed in *Arabidopsis*, requires the presence of chorismate mutase activity in the cytosol, in addition to the activities of prephenate dehydratase and phenylpyruvate aminotransferase. Interestingly, the article by Qian et al. 2019 [32], which describes the aforementioned cytosolic pathway, specifically points to the non-existence of prephenate dehydratase activity in the cytosol in *Pinus pinaster*. Taken together, these data strongly indicate that conifers, and likely all gymnosperms, lack the capacity for cytosolic Phe synthesis. Furthermore, CMs from most ancestral land plants, such as mosses, liverworts, ferns, or lycophytes, apparently lack cytosolic CMs, with the striking exception of *Selaginella moellendorffii*, suggesting that the cytosolic Phe synthesis pathway is also absent in the non-seed plants which first colonized the land. The absence of this pathway also implies a lower capacity for the interconversion between Phe and Tyr in the cytosol, which has been suggested to be important for the catabolism of these amino acids [33]. However, the lineages of plants apparently lacking the cytosolic pathway coincide with those having Phe-hydroxylase activity in the cytosol, thus allowing Phe/Tyr interconversion [34].

Taken together, our work and the literature suggest that the existence of cytosolic CMs correlates with the cytosolic synthesis of Phe, a novel feature of angiosperms. To delve into the evolutionary appearance of this route, we identified and analyzed the sequences of CM enzymes corresponding to basal groups of angiosperms. In these analyses, we determined that *Amborella trichopoda*, the most primitive of existing angiosperms, like conifers, has two plastidial CMs, but lacks cytosolic isoforms. However, the existence of cytosolic CM enzymes in other basal angiosperms considered less primitive, such as *Nymphaea colorata* or *Liriodendron tulipifera,* allows us to delimit the evolutionary point at which the cytosolic synthesis of Phe may arise.

Through phylogenetic analysis, we have shown that CMs from seed plants are distributed in two clades (Figure 1). Clade-II includes allosterically-regulated plastidic CMs from angiosperms and gymnosperms, such as PpCM2, as well as the complete set of CMs from ancient plant lineages such as mosses, ferns, or lycophytes. Clade-III mainly includes cytosolic isoforms of angiosperms, which lack regulation by amino acids [3] and by a group of gymnosperm plastidic CMs, including PpCM1. 

We have demonstrated that PpCM1 and PpCM2 are activated by Trp and inhibited by Phe and Tyr, indicating that allosteric regulation is not exclusive to CMs from clade-II. This result is not surprising, considering that both pine CMs retain the residue Gly213, as described by Westfall et al. (2014) [3] in AtCM1, which is decisive for conferring allosteric sensitivity. In contrast, the occurrence of deregulated enzymes has been associated with the loss of the Gly213. These data suggest that the appearance of seed plants was followed by the emergence of a new group of CMs that, in the case of angiosperms, subsequently gave rise to the current deregulated cytosolic enzymes, requiring the loss of both, the signal peptide for plastid targeting, and the residue Gly213. 

To delve deeper into this point, we checked the presence of Gly213 in the set of sequences used to build the phylogenetic tree. While in most of these sequences, there is a concurrent loss of both plastidial localization and Gly213, we observed that members of the paraphyletic group of basal eudicots, represented here by *Aquilegia coerulea* and *Eschscholzia californica,* lack plastid signal peptide but retain Gly213, likely preserving their allosteric sensitivity. The analysis of this plants group suggests that during plant evolution, some CM enzymes lost their plastid localization and, subsequently, their allosteric sensitivity. A possible interpretation could be that the loss of this regulation could be associated with the cytosolic localization, where competition with anthranilate synthase for the synthesis of Trp does not occur. On the other hand, the levels of Phe and/or Tyr would predictably be lower, due to lower synthesis and immediate consumption by various cytosolic routes, such as the synthesis of phenylpropanoids.

The existence of two CM isoforms in pine, as observed in other gymnosperms, suggests the existence of alternative and likely non-overlapping physiological roles. Our expression analyzes show that in pine seedlings, the two isoforms present markedly different expression patterns. PpCM1 expression is preferential in cotyledon mesophyll cells, hypocotyl pith, and in the vascular tissues of developing roots, while PpCM2 is preferentially expressed in vascular tissues from young needles, vascular cells of cotyledons, and hypocotyls. We also analyzed the expression of PpCM1 and PpCM2 in the highly lignified compression wood (CW) versus opposite wood (OW), which is characterized by lower levels of this polymer. Although the level of expression of both isoforms in CW is similar, the induction of PpCM2, compared to OW, (5.2x) is much higher than in PpCM1 (1.4X). The induction of PpCM2 could be associated with a higher rate of Phe synthesis to support the massive synthesis of lignin, distinctive of CW, but also of the vascular tissues of seedlings. Supporting this hypothesis, PpCM2 is more resistant to Phe inhibition than PpCM1, and retains considerable activity at high levels of Phe (30–35% at 2 mM), enabling this enzyme to generate a higher rate of Phe synthesis for lignification, consistent with the presence of two Phe-insensitive type ADTs, such as PpADT-A and PpADT-D [11], in the same tissue. This hypothesis is also supported by our observation in previous investigations that the expression of *PpCM2*, but not PpCM1, is significantly reduced in plants silenced for *PpMYB8*, a central transcription factor in the coordination of Phe and lignin synthesis and a transcriptional activator of *PpADT-A* and *PpADT-D* [14]. Furthermore, our co-immunoprecipitation analysis revealed the in vivo formation of heterodimers between PpCM1 and PpCM2, indicating a potential functional interaction between these isoforms. This finding suggests a cooperative role for these isoforms in certain physiological contexts, possibly contributing to the fine-tuning of Phe synthesis and metabolic regulation within plastids. While the formation of heterodimers among CM isoforms has not been previously reported, this discovery opens new avenues for understanding the molecular mechanisms underlying metabolic regulation in plants, particularly in the context of aromatic amino acid biosynthesis.

Although the role of the Phe-cytosolic pathway in angiosperms remains unclear, its involvement in plant biotic and abiotic stress responses has been proposed [32]. The non-existence of this route in gymnosperms and other more ancestral groups of plants, suggests that its evolutionary appearance would be related to specific environmental adaptation mechanisms developed later along the evolution of angiosperms.

Finally, our results demonstrate that although pine CM activation by Trp is limited (3.1-fold for PpCM1 and 1.7-fold for PpCM2), the EC_50_ values indicate that these enzymes show extraordinary sensitivity towards this effector, compared to previously characterized enzymes from other plants. To fully understand the importance of this aspect, future work, focused on the allosteric regulation of the pine anthranilate synthase enzyme, is necessary.

## 5. Conclusions

The studies on the Phe biosynthetic pathway in *Pinus pinaster* provide new insights into the evolution and regulation of chorismate mutase (CM) enzymes across plant species. We identified two CM isoforms, PpCM1 and PpCM2, specifically located in the plastid, suggesting the absence of cytosolic CMs in conifers. These isoforms exhibit differential expression patterns supporting distinct roles, with PpCM2 showing higher induction levels and resistance to Phe inhibition compared to PpCM1. The formation of heterodimers between PpCM1 and PpCM2 indicate a potential cooperative regulation of Phe synthesis within plastids, at least under certain physiological and developmental conditions. Moreover, the absence of cytosolic Phe synthesis pathways in gymnosperms underscores the unique evolutionary adaptations in angiosperms. Our findings highlight the remarkable sensitivity of pine CMs to tryptophan (Trp) activation, emphasizing the crucial role of plastid-localized CMs in modulating aromatic amino acid metabolism.

## Figures and Tables

**Figure 1 cells-13-00929-f001:**
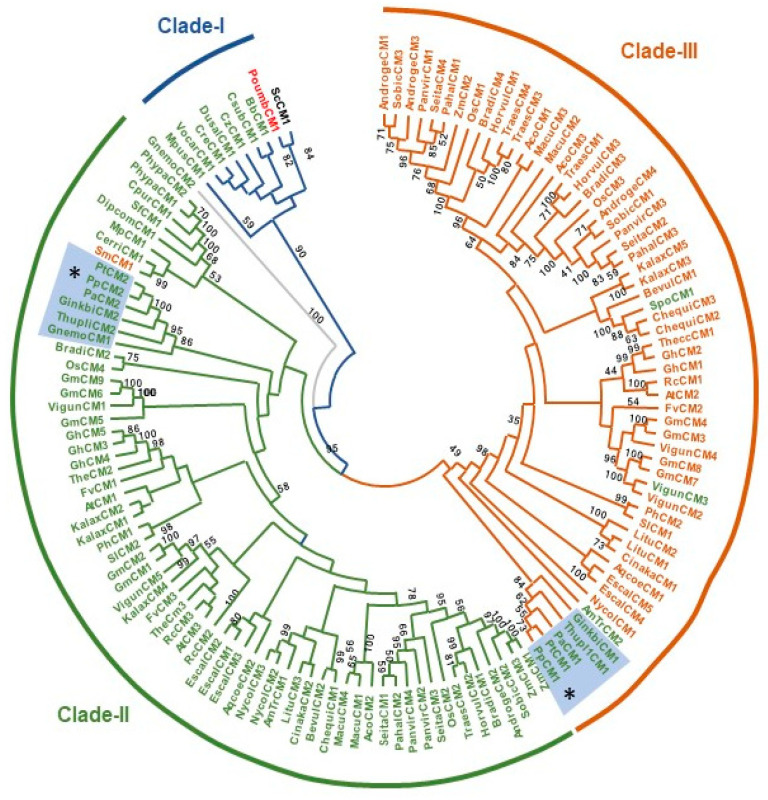
Maximum-likelihood phylogeny of plant chorismate mutase (CM) proteins. Phylogeny of the 135 plant CM protein sequences inferred using maximum likelihood. Sequences were aligned using MUSCLE [15] and the tree was produced using PhyML. Bootstrapping was performed with 1000 replicates. Abbreviations: AmTr, *Amborella trichopoda*; Aco, *Ananas comosus*; Aqcoe, *Aquilegia coerulea*; At, *Arabidopsis thaliana*; Bb, *Botryococcus braunii*; Bradi, *Brachypodium distachyon*; Cpur, *Ceratodon purpureus*; Chequi, *Chenopodium quinoa*; Cre, *Chlamydomonas reinhardtii*; Cz, *Chromochloris zofingiensis*; Cerri, *Ceratopteris richardii*; Cinaka, *Cinnamomum kanehirae*; Csub, *Coccomyxa subellipsoidea*; Dipcom, *Diphasiastrum complanatum*; Dusal, *Dunaliella salina*; Fv, *Fragaria vesca*; Gm, *Glycine max*; Gh, *Gossypium hirsutum*; Litu, *Liriodendron tulipifera*; Mp, *Marchantia polymorpha*; Mpus, *Micromonas pusilla*; Macu, *Musa acuminata*; Nycol, *Nymphaea colorata*; Os, *Oryza sativa*; Pahal, *Panicum hallii*; Ph, *Petunia hybrida*; Phypa, *Physcomitrella patens*; Pa, *Picea abies*; Pp, *Pinus pinaster*; Pt, *Pinus taeda*; Poumb, *Porphyra umbilicalis*; Rc, *Ricinus communis*; Sm, *Selaginella moellendorffii*; Sl, *Solanum lycopersicum*; Sobic, *Sorghum bicolor*; Sf, *Sphagnum fallax*; The, *Theobroma cacao*; Thupli, *Thuja plicata*; Vigun, *Vigna unguiculata*; Vocar, *Volvox carteri*, and Zm, *Zea mays* (Appendix A). Sequences included in the tree were predicted mature peptides, putative chloroplast transit peptides were removed. * Asterisks indicate the phylogenetic position of CM proteins from *Pinus pinaster*.

**Figure 2 cells-13-00929-f002:**
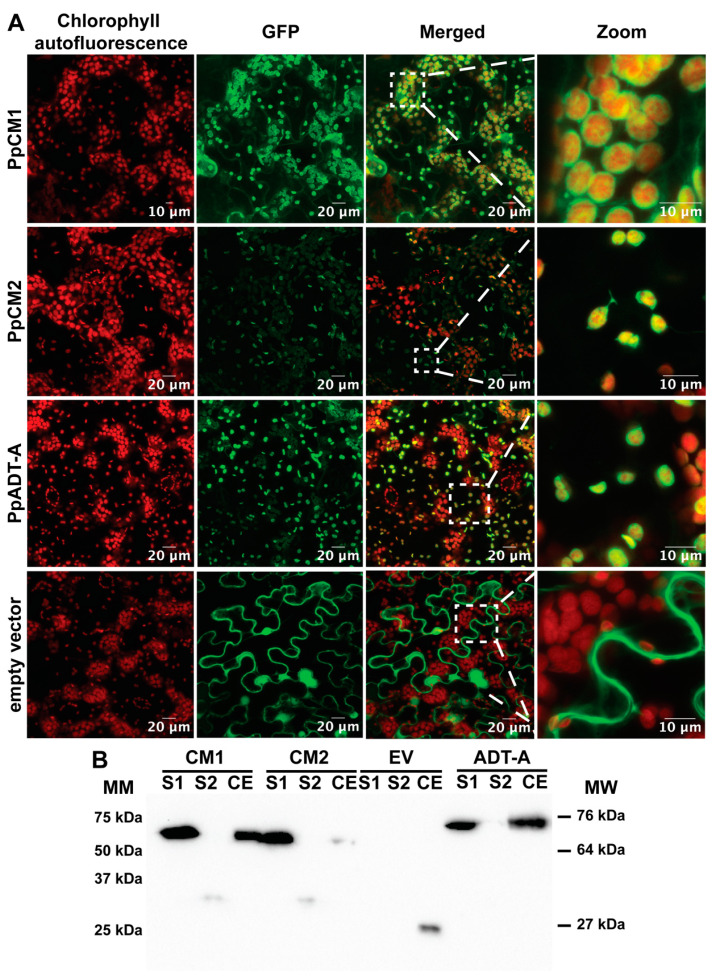
Subcellular localization of PpCM1 and PpCM2 in transient expressed protein in *N. benthamiana* leaves. (**A**) Confocal microscopy images of PpCM1, PpCM2, PpADT-A, and empty vector using chlorophyll autofluorescence as an indicator of chloroplast; expression of PpCMs/PpADT-A/empty vector-GFP, merged images, and zoomed-in view of the chloroplast from left to right. (**B**) Western blot detecting GFP-tag of chloroplast isolation from agroinfiltrated transient *N. benthamiana* disk leaves. Molecular markers (MM), chloroplast isolated soluble proteins (S1), chloroplast isolated insoluble proteins (S2), disk leaf total proteins (CE), molecular weight (MW). CM1 (PpCM1), CM2 (PpCM2), EV (empty vector), ADT-A (PpADT-A). PpCM1,2, PpADT-A as a positive control and pGWB6 empty vector as a negative control. MW for the proteins of interest are CM1 and CM2 at 64 kDa, ADT-A at 76 kDa, and GFP alone at 27 kDa.

**Figure 3 cells-13-00929-f003:**
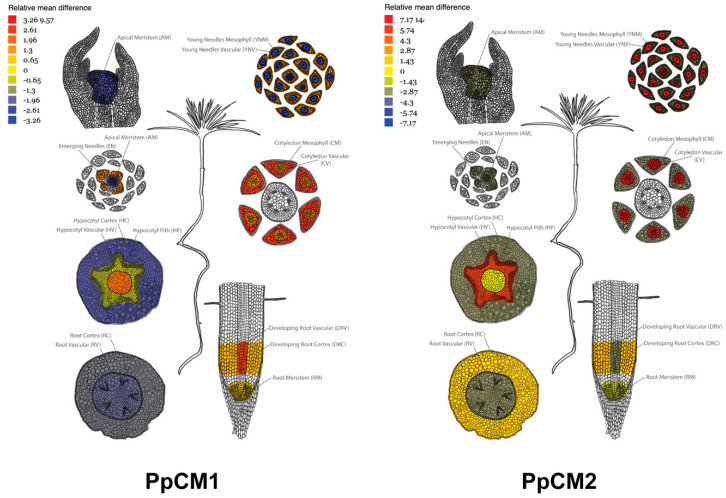
Expression patterns of *PpCM1* and *PpCM2* in pine seedlings. The image was generated using the exImage tool, available at ConGenIE.org (http://v22.popgenie.org/microdisection/, accessed on 15 April 2024), as described by Cañas et al., 2017 [23]. The images depict the expression patterns of *PpCM1* and *PpCM2* in maritime pine seedlings.

**Figure 4 cells-13-00929-f004:**
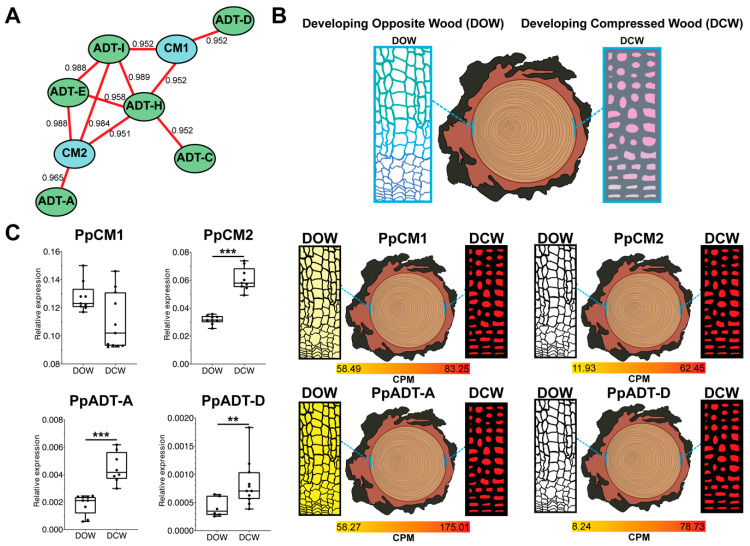
Developing opposite wood (DOW) and developing compressed wood (DCW) expression profile for *PpCMs* and *PpADT* candidates. (**A**) Co-expression network illustrating the correlation between candidate *PpCMs* and *PpADTs*. Red lines signify a positive correlation among interacting genes. (**B**) Expression profiles of *PpCM1*, *PpCM2*, and *PpADTA-D* in DOW and DCW, derived from RNASeq samples, measured in counts per million (CPM). (**C**) Corresponding RT-qPCR results for detecting the expression of candidate genes. Statistically significant differences are indicated with asterisks (** *p* value < 0.05 and *** *p* value < 0.01).

**Figure 5 cells-13-00929-f005:**
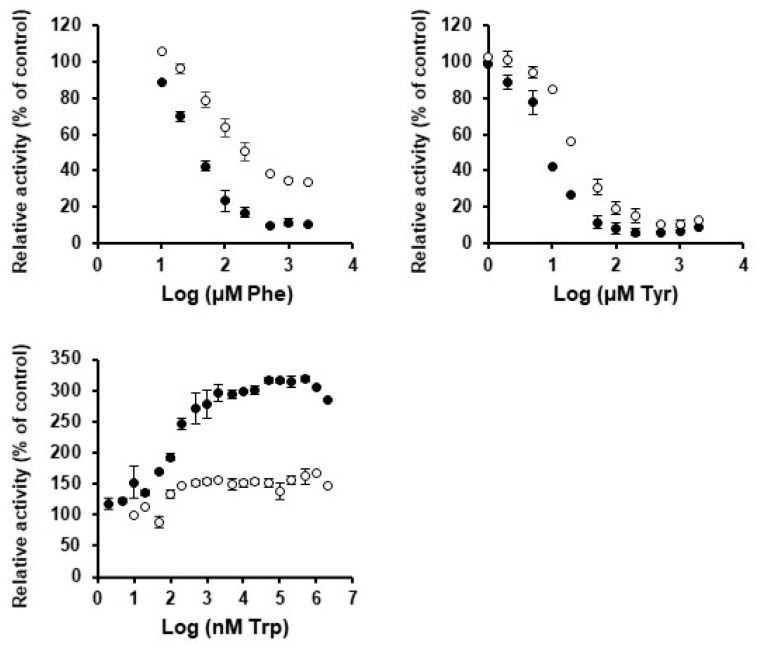
Effect of Phe, Tyr, and Trp on the activity of PpCM1 and PpCM2. Dose-response curve for PpCM1 (solid circles) and PpCM2 (open circles) in the presence of different concentrations of Phe, Tyr, and Trp. Data are the averages ± SE.

**Figure 6 cells-13-00929-f006:**
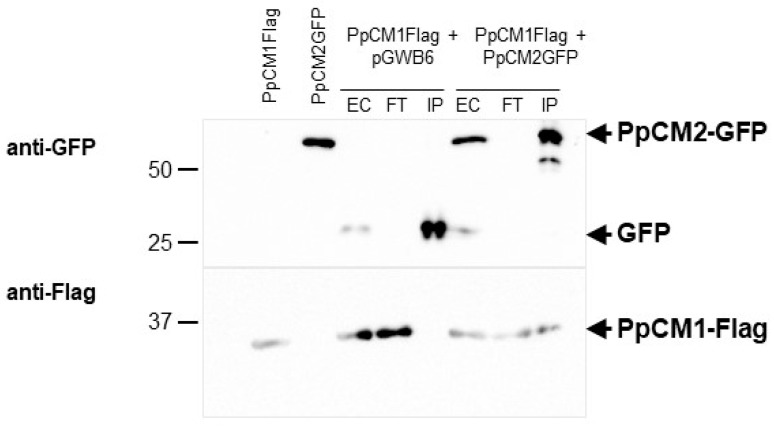
Co-immunoprecipitation assays showing interactions of PpCM1 with PpCM2. Both proteins (PpCM1-Flag and PpCM2-GFP) were transiently expressed in *Nicotiana benthamiana* leaves. Total proteins were immunoprecipitated with GFP-Trap beads, separated by SDS/PAGE and immunoblotting with anti-GFP (upper) and anti-Flag (lower) antibodies. The combination of PpCM1-Flag and pGWB6 was used as a control. Samples corresponding to crude extracts (EC), flow through (FT), and input (IP) are shown.

**Table 1 cells-13-00929-t001:** Kinetic parameters of PpCM isoforms.

Protein	K_m_(mM)	K_cat_ (s^−1^)	K_cat_/K_m_ (s^−1^·mM^−1^)	EC_50_ Tyr (µM)	EC_50_ Tyr (µM)	EC_50_ Tyr (µM)
PpCM1	1.6	29.4	18.4	9.1	36	0.1
PpCM2	1.7	35	20.6	19	88	0.1

## Data Availability

The data presented in this study are available on request from the corresponding author.

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
