# Peer review of "Properties and Functional Analysis of Two Chorismate Mutases from Maritime Pine"

_cells, 2024, doi:10.3390/cells13110929_

Round 1
Reviewer 1 Report
Comments and Suggestions for Authors
This is a well-written study, and I have only few technical points to suggest:
Line 96: Please put a space between „from“ and "Saccharomyces" and italicise Saccharomyces cerevisiae
Line 97: Please put space before “and”
Please make a consistent spelling of °C and consult Instruction for Authors: write it together with the number or number-space-°C
Author Response
Thanks for the comments, we have fixed these mistakes.
All changes are highlighted on the new version of the manuscript. Please see the attachment.

Reviewer 2 Report
Comments and Suggestions for Authors
Summary
This study explored the regulation and evolution of chorismate mutase (CM) enzymes in maritime pine (Pinus pinaster). Two genes, PpCM1 and PpCM2, were identified as encoding plastid-localized CM isoenzymes that differ in function. PpCM2 plays a significant role in phenylalanine and lignin biosynthesis, exhibiting higher expression levels and resistance to phenylalanine inhibition compared to PpCM1. Both enzymes are very sensitive to tryptophan activation. The study also seems to suggest that PpCM1 and PpCM2 can form heterodimers, suggesting potential cooperative regulation of plastid phenylalanine biosynthesis.
Comments to the authors
The paper is well written, and I have only minor comments.
Line 21; Unclear whether ‘activities’ refers to enzyme activity or expression levels.
Line 72; “CM activity” Line 96; “fromSaccharomyces” Line 97; “muscle15and” Line 114; “described by19” Line 115; same as line 114
Line 118; The °C symbol is incorrect
Line 119; Please rephrase, the ratios do not indicate concentration.
Line 121; “through both agarose gel electrophoresis” Line 131; This is a very high cycle number (50), please comment on its necessity. Is the template expected to be found at extremely low concentrations?
Line 136; same as line 114
Line 182; Please explain the reason for this gentle shaking protocol.
Line 192; Please italicise organism name
Line 199; please describe centrifugation conditions
Line 208; Please provide more detail on the antibodies used.
Line 236; “conifers in overall” Line 291; “autofluorescence chlorophyll” Line 311; “performed” Line 313; same as line 114
Line 339; “candidates PpCMs” Line 349; “at A274nm” Line 448; Please confirm that this generalization is valid.
Line 542; same as line 448
Author Response
The paper is well written, and I have only minor comments.
Thanks for the comments
Line 21; Unclear whether ‘activities’ refers to enzyme activity or expression levels. We have changed the sentence to make clear this point.
Line 72; “CM activity” Line 96; “fromSaccharomyces” Line 97; “muscle15and” Line 114; “described by19” Line 115; same as line 114. Fixed
Line 118; The °C symbol is incorrect Fixed
Line 119; Please rephrase, the ratios do not indicate concentration. We have now rewritten the sentence
Line 121; “through both agarose gel electrophoresis” fixed Line 131; This is a very high cycle number (50), please comment on its necessity. Is the template expected to be found at extremely low concentrations? Yes, the expression level of these genes are very low, also most of the genes from the shikimate pathway.
Line 136; same as line 114 Fixed
Line 182; Please explain the reason for this gentle shaking protocol. We already optimized recombinat protein protocols having the best results using this shaking step.
Line 192; Please italicise organism name Fixed
Line 199; please describe centrifugation conditions Now included
Line 208; Please provide more detail on the antibodies used. This information is now included
Line 236; “conifers in overall” Fixed, we have remove "overall" and replaced with globally
Line 291; “autofluorescence chlorophyll” Fixed
Line 311; “performed” Fixed using generated instead of preformed, thanks.
Line 313; same as line 114 Fixed
Line 339; “candidates PpCMs” Fixed, thanks for the indication
Line 349; “at A274nm” Fixed
Line 448; Please confirm that this generalization is valid. We have rewritten the sentence to recognize that only in the case of the CMs of Pinus pinaster we have experimentally determined the subcellular localization. However, analyzes of all identified conifer CM sequences strongly suggest the existence of plastidial localization signals and thus plastidial localization.
Line 542; same as line 448 Through this sentence we suggest that the absence of cytosolic CM in the species Pinus pinaster may be a common characteristic of conifers. This suggestion is reinforced by the fact, described in the work, that all conifer CM proteins analyzed have amino terminal signals that, according to predictions, correspond to plastidial localization peptides.
All changes are highlighted on the new version of the manuscript. Please see the attachment.
